# Synthesis, Characterization and Application of NNN Pincer Manganese Complexes with Pyrazole Framework in α-Alkylation Reaction

**DOI:** 10.3390/molecules30071465

**Published:** 2025-03-26

**Authors:** Tao Wang, Yongli Xu, Mengxin Du, Zhiyuan Hu, Lantao Liu

**Affiliations:** 1Henan Engineering Laboratory of Green Synthesis for Pharmaceuticals, School of Chemistry and Chemical Engineering, Shangqiu Normal University, Shangqiu 476000, China; 2College of Petrochemical Engineering, Liaoning Petrochemical University, Fushun 113001, China

**Keywords:** NNN pincer manganese complexes, pyrazole skeleton, α-alkylation reaction

## Abstract

A series of novel NNN pincer manganese complexes based on pyrazole skeleton **4** were efficiently synthesized in a two-step process. All of the new complexes were fully characterized by ^1^H, ^13^C NMR spectra. Furthermore, the molecular structures of complexes **4a** and **4c** were also determined by X-ray single-crystal diffraction. The manganese(I) catalysts obtained showed efficient catalytic activity in the α-alkylation reaction of ketones with alcohols. Under optimal reaction conditions, the expected products were procured with moderate to high yields.

## 1. Introduction

Over the past few decades, the synthesis of pincer complexes and their applications in catalysis have been significant research topics in the field of organometallic chemistry [1,2,3,4,5,6,7,8]. To date, a wide array of pincer complexes, including those of palladium [9,10,11,12,13], ruthenium [14,15,16,17,18,19], copper [20,21,22], nickel [23,24,25], cobalt [26,27,28,29,30], and zinc [31,32], have been successfully synthesized and utilized as catalysts in numerous challenging chemical reactions. Despite these advancements, the design, synthesis, and application of pincer manganese complexes remain in their infancy. In fact, after iron and titanium, manganese is the third most abundant transition metal in the Earth’s crust. Since the first report of a manganese pincer catalyst in 2016 by the group of Milstein [33], pincer manganese complexes have garnered the attention of many scientists. For instance, the corresponding PNP-manganese pincer complexes [34,35,36,37,38] have been applied to the Guerbet reaction of ethanol, C-alkylation of secondary alcohols, dehydrogenative coupling, and conjugate addition reactions. A novel type of PNN pincer ligand and its corresponding manganese complexes [39,40,41,42,43] have been developed and further applied in the transfer hydrogenation of esters, and hydrogenation reactions, as well as N-alkylation of amines. Due to the well-documented air and moisture sensitivities and the high cost of phosphine ligands, the development of phosphine-free pincer manganese complexes has been reported successively [44,45,46,47]. Despite considerable progress, the construction of new functional organic compounds catalyzed by known metal pincer complexes, as well as the synthesis and applications of new pincer complexes, will continue to be significant contributions to the field of pincer chemistry. Recently, we reported a simplified method for synthesizing pincer palladium complexes [48,49,50] and NHC-palladium complexes [51,52,53,54] featuring tridentate N-heterocyclic ligands. The palladium complexes obtained exhibited outstanding catalytic activity in various catalytic reactions. To expand the scope of pincer metal chemistry and in continuation of our investigations into the construction of functionalized complexes [55,56], we would like to describe the NNN pincer manganese complexes based on a pyrazole skeleton, which can also be conveniently synthesized using 2-pyridinecarboxylic acid and 2-pyrazolyl anilines as starting materials (Figure 1).

## 2. Results and Discussion

### 2.1. Synthesis and Characterization of the Complexes

The synthesis of the required ligands **3** was easily carried out from commercially available 2-pyridinecarboxylic acid, as shown in Figure 1. With the ligands in hand, the coordination was carried out by heating the ligands **3a**–**c** with Mn(CO)_5_Br in toluene in the presence of K_2_CO_3_ to afford the air- and moisture-stable pincer manganese complexes **4a**–**c** in good yields. All of the new complexes were fully characterized by ^1^H NMR and ^13^C NMR. Additionally, the X-ray crystal structures of complex **4a** and **4c** were obtained.

The molecular structures of pincer manganese complexes **4a** and **4c** were unambiguously determined by X-ray single-crystal analysis. The molecules are illustrated in Figure 1. The bond lengths and bond angles selected are listed in Table 1. The molecular structure of each complex exhibits an octahedral geometry with the pincer ligand coordinated to the metal via the pendant pyridine-N, amide-N, and pyrazol-N in tridentate manner and three carbonyl ligands occupying the other coordination sites. Complexes **4a** and **4c** possess one five-membered and one six-membered manganese ring. All of the bond lengths and angles around the Mn(I) center in the two complexes are similar, and the N(amide)-Mn-C angles (174°) are almost linear. The values of bond lengths and angles are comparable to those in the related pincer manganese complexes [57]. In complex **4a**, the Mn-N bond lengths are 2.074(4) Å, 2.003(3) Å and 2.076(4) Å, respectively. The N(1)-Mn(1)-N(2), N(1)-Mn(1)-N(4) and N(2)-Mn(1)-N(4) angles in this complex are 77.65(14)°, 92.83° and 80.10(14)°, respectively. Complex **4c** has very similar bond lengths and angles around the Mn(I) center, although it contains a different substituent on its pyrazol ring from **4a**.

### 2.2. Catalytic Studies

The α-alkylation reaction of carbonyl compounds offers a valuable method for constructing a range of functionalized organic molecules, which exhibit activities in pharmaceuticals, materials science, and agricultural chemicals [19]. Recently, several research groups have achieved favorable outcomes in α-alkylation reactions by employing synthetic cobalt [58], ruthenium [59,60], and zinc [61] as catalysts. Furthermore, an excess of base can also facilitate the occurrence of the α-alkylation reaction [62]. However, the aforementioned literature reports still present numerous issues, such as complex catalyst synthesis processes and excessive amounts of alkali used. Therefore, it is imperative to develop more effective strategies for the α-alkylation reaction. In this paper, the α-alkylation reaction of acetophenone **5a** with 4-methoxybenzyl alcohol **6a**, using the current NNN pincer manganese complex **4c** as a catalyst and KO^t^Bu as the base, was conducted in toluene at 140 °C for 14 h. To our satisfaction, the desired product **7aa** was produced with a 35% yield (Table 2, entry 1). Various other bases, including KOH, K_2_CO_3_, Na_2_CO_3_, NaO^t^Bu, NaOH, LiCO_3_, LiO^t^Bu, Cs_2_CO_3_, and Et_3_N, were subsequently tested (Table 3, entries 2–10). Notably, NaOH emerged as the superior base, yielding the product at 81% (entry 6). This underscores the importance of the base selection on the reaction’s yield [63]. After exploring various solvents, including 1,4-Dioxane, m-xylene, tert-amyl alcohol, THF, and DMF (entries 11–15), it was found that none of these solvents improved the product yield. Finally, the effectiveness of two other NNN pincer manganese complexes **4a**–**4b** under these reaction conditions was evaluated, and complex **4c** was determined to be the most effective (Table 2, entries 16–17 compared to entry 6). The product yield was determined to be 57% under reaction conditions with the temperature set at 120 °C (entry 18). When 2.0 mol% of complex **4c** was tested, the yield of the product decreased significantly (entry 19).

With the optimized conditions at our disposal, we began to systematically explore the generality of this methodology. Various alcohols were tested for alkylation with **5a**. As presented in Table 3, the majority of the α-alkylation reactions proceeded efficiently, yielding the corresponding products **7aa**–**7ah** with good to excellent results. Both electron-donating and electron-withdrawing groups on the phenyl ring of the aryl alcohols were tolerated (**7aa**–**7af**). Generally, the presence of electron-donating groups in the phenyl ring of aryl alcohols had a beneficial effect on the yields of the catalytic products. However, the presence of substituents at the ortho or para positions of benzyl alcohol could lead to a reduction in product yields (**7ab** and **7ac**). We attempted to use 4-nitrobenzyl alcohol and 4-trifluoromethylbenzyl alcohol as substrates for the reaction, but, unfortunately, the target compound was not obtained. Additionally, substrate **6**, which contains a naphthalene ring substituent, was also suitable for such transformation, producing products **7ag** and **7ah** in good yields under appropriate conditions.

Subsequently, we examined the scope of the ketone component (Table 4). A broad spectrum of ketones incorporating diverse functional groups were well tolerated, resulting in the desired products **7ba**–**7ha** with good yields. Overall, the current findings indicated that the NNN pincer manganese complex **4c** remained efficient in the α-alkylation reaction process.

## 3. Experimental Section

### 3.1. General

The preparation of the NNN pincer manganese complexes and all catalytic reactions were conducted under a nitrogen atmosphere. Ligand **3** was synthesized following the procedure described in the literature [48]. Solvents were dried using standard methods and distilled freshly before use. All other chemicals were used as purchased. Melting points were determined using an XT4A melting point apparatus and are uncorrected. ^1^H and ^13^C NMR spectra were obtained on a Germany Bruker DPX 400 instrument, with TMS serving as the internal standard.

### 3.2. General Procedure for Synthesis of the NNN Pincer Manganese(I) Complexes 4

To a stirred solution of **3** (0.20 mmol) and K_2_CO_3_ (0.30 mmol) in THF (2 mL), Mn(CO)_5_Br (0.30 mmol) was added under a N_2_ atmosphere, and the reaction mixture was reacted for 12 h. After cooling, filtration, and evaporation, the residue was purified by preparative TLC on silica gel plates to afford the corresponding NNN pincer manganese(I) complexes **4a**–**c**.

**Complex** (**4a**): 77.4 mg, 90% yield. Yellow solid. mp: 220–223 °C. ^1^H NMR (400 MHz, CDCl_3_) δ 9.02 (d, *J* = 3.8 Hz, 1H), 8.05 (d, *J* = 6.4 Hz, 1H), 7.90 (s, br, 1H), 7.65 (d, *J* = 6.4 Hz, 1H), 7.46 (d, *J* = 5.9 Hz, 2H), 7.26 (s, br, 2H), 5.95 (s, 1H), 2.36 (s, br, 6H). ^13^C NMR (100 MHz, CDCl_3_): δ 166.7, 157.3, 152.7, 151.9, 146.8, 142.9, 138.5, 133.1, 128.6, 126.1, 125.8, 125.2, 123.8, 123.5, 109.8, 15.2, 14.5.

**Complex** (**4b**): 81.5 mg, 89% yield. Yellow solid. mp: 228–230 °C. ^1^H NMR (400 MHz, CDCl_3_) δ 9.00 (d, *J* = 5.1 Hz, 1H), 8.04 (d, *J* = 7.2 Hz, 1H), 7.89 (t, *J* = 7.2 Hz, 1H), 7.65 (d, *J* = 7.5 Hz, 1H), 7.48–7.43 (m, 2H), 7.28 (d, *J* = 7.6 Hz, 1H), 7.21 (t, *J* = 7.6 Hz, 1H), 6.04 (s, 1H), 3.08–2.98 (m, 1H), 2.88–2.80 (m, 1H), 2.66–2.57 (m, 2H), 1.18 (t, *J* = 7.4 Hz, 3H), 1.08 (t, *J* = 7.5 Hz, 3H). ^13^C NMR (100 MHz, CDCl_3_): δ 166.4, 157.8, 157.4, 152.7, 149.4, 147.1, 138.4, 133.1, 128.5, 126.0, 125.9, 125.1, 123.7, 123.6, 105.5, 22.5, 21.6, 13.4, 12.7.

**Complex (4c)**: 90.5 mg, 92% yield. Yellow solid. mp: 234–236 °C. ^1^H NMR (400 MHz, CDCl_3_) δ 9.05 (s, 1H), 8.08 (s, br, 1H), 7.92 (d, br, 1H), 7.63 (d, *J* = 6.0 Hz, 1H), 7.49 (s, br, 1H), 7.35–7.26 (m, 4H), 7.20 (s, 2H), 6.86 (s, br, 1H), 6.66 (d, *J* = 6.8 Hz, 1H), 6.17 (s, 1H), 2.48 (s, 3H). ^13^C NMR (100 MHz, CDCl_3_): δ 153.1, 152.8, 147.6, 146.9, 138.7, 133.2, 130.1, 129.2, 129.1, 128.7, 128.5, 126.1, 126.0, 125.9, 125.4, 123.7, 109.9, 15.6.

### 3.3. General Procedure for the α-Alkylation Reaction

A Schlenk flask was charged with the required ketone (0.50 mmol), alcohol (0.75 mmol), NNN pincer manganese(I) complex (5.0 mol %), NaOH (50.0 mol %), and toluene (2.0 mL). The mixture was stirred at 140 °C for 14 h under N_2_. After cooling, the reaction mixture was evaporated and the product was isolated by preparative TLC on silica gel plates. The purified products were identified by NMR spectra and their analytical data are given in the Appendix A.

### 3.4. Crystal Structure Determination and Data Collection

Crystals of complex **4a** (CCDC 2422192) and **4c** (CCDC 2422233) were obtained by recrystallization from CH_2_Cl_2_/*n*-hexane at ambient temperature. Data were collected on an Oxford Diffraction Gemini E diffractometer with graphite-monochromated Mo Kα radiation (λ = 0.7107 Å). The structure was solved by direct methods using the SHELXS-97 program, and all non-hydrogen atoms were refined anisotropically on *F*^2^ by the full-matrix least-squares technique, using the SHELXL-97 crystallographic software package [64,65]. The hydrogen atoms were included but not refined. Details of the crystal structure determination are summarized in Table 5. The crystallographic data can be obtained free of charge from The Cambridge Crystallographic Data Centre via http://www.ccdc.cam.ac.uk/data_request/cif (accessed on 7 December 2024).

## 4. Conclusions

In conclusion, a new series of phosphine-free pincer manganese complexes based on a pyrazole framework have been synthesized and utilized as catalysts for the α-alkylation of ketones with alcohols. Ongoing research in our laboratory focuses on further modifications of these NNN pincer manganese complexes and their application as catalysts in various other reactions.

## Data Availability

The original contributions presented in this study are included in the article/Appendix A. Further inquiries can be directed to the corresponding author.

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
