# Peer review of "Synthesis, Characterization and Application of NNN Pincer Manganese Complexes with Pyrazole Framework in α-Alkylation Reaction"

_molecules, 2025, doi:10.3390/molecules30071465_

Round 1
Reviewer 1 Report
Comments and Suggestions for Authors
The synthesis and characterization of the complex is interesting and worth publication. However, a comparison of the results of the α-alkylation reaction of acetophenone with literature is mandatory since the yields are low to good.
It should be mentioned that high yield have been obtained with Ru (Chemistry - An Asian Journal (2025), 20(1), e202400811, European Journal of Organic Chemistry (2024), 27(24), e202400131), Co (Dalton Transactions (2021), 50(24), 8567-8587), Zn (Chemical Communications, United Kingdom) (2024), 60(30), 4056-4059), … even catalyst free reaction (Angewandte Chemie, International Edition (2014), 53(1), 225-229).
Thus, the authors must compare their results with the literature and must discuss the pros and cons of their catalyst.
Comments on the Quality of English Languageno comment
Author Response
- Comment:
The synthesis and characterization of the complex is interesting and worth publication. However, a comparison of the results of the α-alkylation reaction of acetophenone with literature is mandatory since the yields are low to good. It should be mentioned that high yield have been obtained with Ru (Chemistry - An Asian Journal (2025), 20(1), e202400811, European Journal of Organic Chemistry (2024), 27(24), e202400131), Co (Dalton Transactions (2021), 50(24), 8567-8587), Zn (Chemical Communications, United Kingdom) (2024), 60(30), 4056-4059), … even catalyst free reaction (Angewandte Chemie, International Edition (2014), 53(1), 225-229).
Thus, the authors must compare their results with the literature and must discuss the pros and cons of their catalyst..
Answer:
The corresponding description has been given (see revised manuscript Page 3: row 83-92).
The reference has been cited (see revised manuscript Page 9: row 244-249, References 16-20 ).
Reviewer 2 Report
Comments and Suggestions for Authors
The paper of Wang and co-workers describes preparation and characterisation of Mn complexes with N,N,N pincer ligands and their use in alpha-alkylation of arylmethyl ketones with benzylic alcohols. The Mn-promoted hydrogen-borrowing mechanism represents a valuable approach to the achievement of alkylated ketones, starting from readily available alcohols, but the paper needs to be improved to be publishable in Molecules, mainly as far as the catalytic studies are concerned.
To optimize the reaction conditions the authors performed a screening of base and solvent, but used always the same catalyst loading and the same reaction temperature. Why this choice? Are those already used conditions? The authors have to report a reference about this point or, even better, have to perform a screening of the catalyst loading and of the reaction temperature, reporting the data in the supporting information.
The scope of the reaction was demonstrated with few examples: in particular the authors evaluated the effect of electro-donating groups, such as alkyls and methoxy, but very few example of the effect of electro-withdrawing groups are present. In addition, nitrogen-containing groups, such as nitro and cyano, are lacking. The effect of these groups must be evaluated.
The 13C NMR spectra of compounds 7ae 7ca 7da 7ba show the presence of more signals than those allied to the compound, in particular two signals in the frequency region of the carbonyl group. Please purify the products and re-consider the reported yield.
Other minor revisions: row 32: use capital letter for Guerbet; row 62: an octahedral; row 63: tridentate
Author Response
(1) Comment:
The paper of Wang and co-workers describes preparation and characterisation of Mn complexes with N,N,N pincer ligands and their use in alpha-alkylation of arylmethyl ketones with benzylic alcohols. The Mn-promoted hydrogen-borrowing mechanism represents a valuable approach to the achievement of alkylated ketones, starting from readily available alcohols, but the paper needs to be improved to be publishable in Molecules, mainly as far as the catalytic studies are concerned.
To optimize the reaction conditions the authors performed a screening of base and solvent, but used always the same catalyst loading and the same reaction temperature. Why this choice? Are those already used conditions? The authors have to report a reference about this point or, even better, have to perform a screening of the catalyst loading and of the reaction temperature, reporting the data in the supporting information.
Answer:
The corresponding content has been modified. The product yield was determined to be 57% under reaction conditions with temperature set at 120 oC. When 2.0 mol% of complex 4c was tested, the yield of the product decreased significantly.(see revised manuscript Page 4: row 104-106, Table 2(entries 18-19).)
(2) Comment:
The scope of the reaction was demonstrated with few examples: in particular the authors evaluated the effect of electro-donating groups, such as alkyls and methoxy, but very few example of the effect of electro-withdrawing groups are present. In addition, nitrogen-containing groups, such as nitro and cyano, are lacking. The effect of these groups must be evaluated.
Answer:
In Table 3, we attempted to use 4-nitrobenzyl alcohol and 4-trifluoromethylbenzyl alcohol as substrates for the reaction, but unfortunately, we were unable to obtain the target compound. In Table 4, 4-chloroacetophenone and 4-trifluoromethylacetophenone showed lower yields. In summary, connecting electron donating groups to the substrate is advantageous for this reaction.
(3) Comment:
The 13C NMR spectra of compounds 7ae 7ca 7da 7ba show the presence of more signals than those allied to the compound, in particular two signals in the frequency region of the carbonyl group. Please purify the products and re-consider the reported yield.
Answer:
The characterization data and 13C NMR spectra of compounds 7ae 7ca 7da 7ba have been revised (see revised manuscript Page 5-6, Table 3-4; revised supporting information, Pages S5-6, S12, S16-18).
(4) Comment:
Other minor revisions: row 32: use capital letter for Guerbet; row 62: an octahedral; row 63: tridentate
Answer:
Other minor revisions have been revised (see revised manuscript Page 1: row 32, Page 2: row 65 \67).
Round 2
Reviewer 2 Report
Comments and Suggestions for Authors
The authors have to discuss failure of the reaction with substrates endowed with nitro groups and underline that the method can work only if substrate possess electrodonating groups
Author Response
Comment:
The authors have to discuss failure of the reaction with substrates endowed with nitro groups and underline that the method can work only if substrate possess electrodonating groups..
Answer:
The corresponding description has been given (see revised manuscript Page 4: row 120-122).